# Markovian Transformers for Informative Language Modeling

**Scott W. Viteri, Max Lamparth, Peter Chatain & Clark Barrett**
Department of Computer Science
Stanford University
Stanford, CA 94305, USA
`{sviteri,lamparth,pchatain,barrettc}@stanford.edu`

## Abstract

Chain-of-Thought (CoT) reasoning often fails to faithfully reflect a language model's underlying decision process. We address this by introducing a *Markovian* language model framework with an autoencoder-style *reasoning bottleneck*: all information flowing from question to answer must pass through a bounded-length CoT, creating a bandwidth bottleneck analogous to the latent layer of an autoencoder. In practice, the KL penalty toward the pretrained distribution and the inductive biases of gradient descent discourage steganographic encoding, so the model learns to express its reasoning in natural-language steps from which the answer can be derived. We train this system with a GRPO-style policy gradient algorithm using parallel sampling, a frozen baseline $CoT'$, within-batch standardized advantages, and actor-reward (chain-rule) gradients. On QA tasks, Markovian training recovers most of the gains of a Non-Markovian GRPO variant while forcing the model to answer from the CoT alone (e.g., GSM8K: 19.6% $\rightarrow$ 57.1%; ARC-Challenge: 36.1% $\rightarrow$ 79.9%; on average within $\approx$3–4 pp of a Non-Markovian variant). Perturbation analyses across types and severities show that Markovian models incur systematically larger log-probability drops under CoT corruption than matched Non-Markovian baselines, indicating stronger causal reliance on the CoT. Cross-model evaluation confirms that learned CoTs generalize across architectures, suggesting they encode transferable reasoning steps rather than model-specific artifacts.

## 1 Introduction

The rapid advancement of language models (LMs) has led to impressive performance on complex cognitive tasks (Brown et al., 2020). Yet it is often unclear *why* an LM arrives at a particular conclusion (Lamparth & Reuel, 2024; Burns et al., 2023; Gurnee & Tegmark, 2024), causing issues in high-stakes applications (Grabb et al., 2024; Lamparth et al., 2025; Rivera et al., 2024). Traditional interpretability methods analyze hidden activations or attention patterns to extract "explanations" (Geiger et al., 2022; Geva et al., 2022; Meng et al., 2022; Casper et al., 2023; Wang et al., 2023; Lamparth & Reuel, 2024; Nanda et al., 2023). Modern LMs, however, already generate coherent text: we might hope *prompting* the model to articulate its reasoning ("Chain-of-Thought" or CoT) (Nye et al., 2022; Wei et al., 2022) would yield a faithful record of its thought process.

Unfortunately, CoT explanations can be *unfaithful*. For example, Turpin et al. (2023) show that spurious in-context biases often remain hidden in the CoT, and Lanham et al. (2023) find that altering CoT text may not affect the final answer. Such observations indicate that standard CoTs are not "load-bearing."

In this work, we take a *pragmatic* approach to interpretability, focusing on *informativeness* over full faithfulness. Rather than insisting the CoT mirrors the model's entire internal process, we require that *the CoT alone suffices to produce the final answer*. In other words, if we remove the original prompt and rely only on the CoT, the model should still reach the correct output. This makes the CoT *causally essential* and *fragile*: perturbing it should reduce answer quality.

What distinguishes our approach is a *structural* constraint rather than a purely optimization-based one (Figure 1). Traditional approaches train models to generate better-quality CoTs, but because the model can still access the original question when producing its answer, nothing prevents it from bypassing the CoT entirely. Our Markovian framework removes this architectural escape hatch: the answer must be predicted from the CoT alone, creating a bandwidth bottleneck through which all relevant information must flow. Like an autoencoder with a narrow latent layer, this forces the model to compress essential reasoning into the CoT, making it *causally load-bearing* for prediction.

For instance, Llama's CoT on arithmetic tasks changed dramatically after training. **Before training**, it simply listed all numbers and their (incorrect) sum (e.g., "Sum = 76 + 90 + 92 + ... = 2314"). **After training**, it performed correct step-by-step calculations (e.g., "calculate 6 + 89 = 95; Next, calculate 95 + 38 = 133..."), breaking the task into manageable steps that can be verified independently and enabling accurate answer prediction even when the original question is removed.

**Autoencoder analogy, time-bounded complexity, and cross-model generalization.** Each token is produced by a single forward pass through a fixed-depth network. For hard problems, the model cannot solve the problem during the $|A|$ forward passes spent reading the question (cf. time-bounded complexity (Li & Vitányi, 2019)). The CoT provides $|B|$ additional forward passes in which to work through the reasoning, and the Markovian constraint channels all information through a $|B|$-token bottleneck, forcing the model to distill into $B$ whatever reasoning about $A$ is needed to predict $C$—analogous to the compression imposed by an autoencoder's narrow latent layer, though here the goal is prediction rather than reconstruction. In principle, this bottleneck could be circumvented via steganography: an unbounded encoder could compute $C$ and store it in $B$ directly, while an unbounded decoder would let the encoder hide $A$ in $B$ via an unnatural bijection. In practice, the KL penalty toward the pretrained distribution and the inductive biases of gradient descent discourage such encodings, making natural-language reasoning the easier solution to discover. Our cross-model experiments (Section 5.4) provide empirical confirmation: CoTs generated by Llama transfer to Mistral, Phi, and even GPT-2—which cannot decode sophisticated steganography—indicating that the learned CoTs encode reasoning in natural language rather than model-specific artifacts. In Section 3.4 we give a minimum description length interpretation of the loss components.

**Contributions.**

1. We introduce a Markovian language model framework that structurally enforces CoT generation to be causally essential, together with a GRPO-style training recipe (parallel sampling, frozen CoT baseline, actor-reward gradients) that optimizes this objective through a discrete text bottleneck.

2. We apply this framework to arithmetic problems and standard QA datasets (GSM8K, MMLU, SVAMP, ARC-Challenge) using Llama 3.1 8B (Grattafiori et al., 2024) and Mistral 7B (Jiang et al., 2023), observing large absolute gains over the base model (e.g., GSM8K $19.6\% \to 57.1\%$, ARC-Challenge $36.1\% \to 79.9\%$) while remaining within $\approx$3–4 percentage points of a Non-Markovian GRPO variant that can still see the question during answer prediction.

3. We show through systematic perturbation analyses on Wikipedia continuation and multiple QA datasets that Markovian training produces consistently higher sensitivity to CoT perturbations compared to matched Non-Markovian baselines (Tables 1 and 2), indicating that the learned CoTs are more causally load-bearing.

4. We demonstrate cross-model transfer: on GSM8K, CoTs generated by Llama 3.1 8B remain informative for diverse other models (Mistral, Phi, GPT-2); on Wikipedia, CoTs generated by Mistral transfer to Llama as evaluator. Transfer even to GPT-2, which cannot decode sophisticated steganography, confirms that the CoTs encode reasoning in natural language.

Section 2 reviews related work, Section 3 details our Markovian framework, and Section 4 describes the RL training. Section 5 presents empirical results, and Section 6 discusses limitations and future directions.

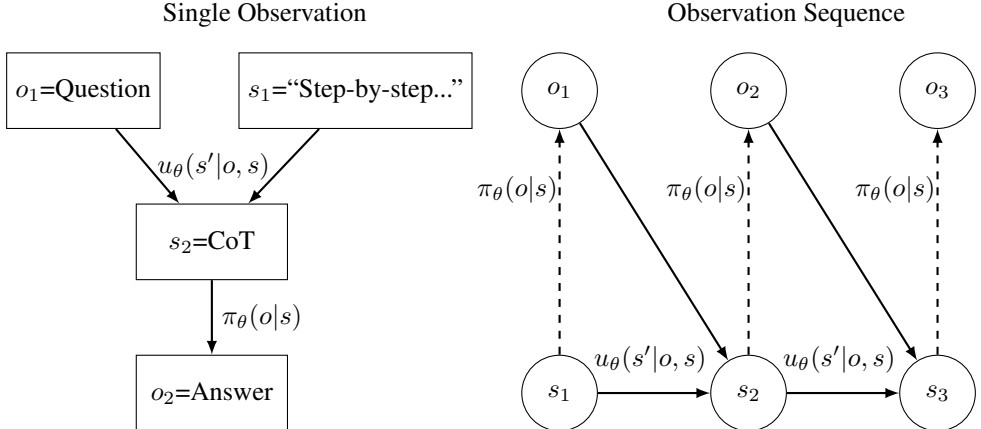

Figure 1: Markovian training as an autoencoder-style reasoning bottleneck. **Left:** Single time-step process from Question to CoT to Answer, creating a bandwidth bottleneck where the CoT must capture all information needed for answer prediction. **Right:** Causal structure showing the generation of states from observations and previous states using the state update function $u_\theta(s'|o, s)$, and the prediction of observations from states using the policy $\pi_\theta(o|s)$. The discrete text bottleneck prevents direct backpropagation, necessitating RL-based gradient estimation.

| Dataset | Baseline | Non-Mkv | Mkv |
|---|---|---|---|
| GSM8K | 19.6% | 63.3% | 57.1% |
| ARC-Chal | 36.1% | 78.6% | 79.9% |
| Arithmetic | 1.0% | 97.0% | 98.0% |
| MMLU | 21.4% | 68.7% | 55.5% |
| SVAMP | 18.0% | 43.3% | 42.3% |

Figure 2: **Left: Accuracy comparison.** Markovian models (Mkv) maintain competitive performance with Non-Markovian counterparts despite the strict information bottleneck. **Right: Wiki perturbation sensitivity** (positive = Mkv more fragile). Markovian models are consistently more sensitive to CoT corruption (higher $\Delta \ln P$), confirming the CoT is causally load-bearing.

## 2 RELATED WORK

Prior work shows that CoT prompting can boost performance on reasoning tasks (Wei et al., 2022; Nye et al., 2022). Whereas typical CoT prompting methods do not alter a pre-trained model's parameters, some prior approaches do fine-tune the model for CoT generation (Zelikman et al., 2022; 2024; Guo et al., 2025). Our work differs by removing the original question or passage from the answer-prediction context, which enforces a stronger causal reliance on the CoT.

Regarding faithfulness vs. interpretability, some authors discuss how a CoT may fail to reflect the true reason the LM arrived at its answer (Lanham et al., 2023; Turpin et al., 2023), since small changes in the CoT do not necessarily change the final prediction. Ton et al. (2025) analyze CoT through an information-theoretic lens, finding that CoT can serve as a communication channel between different parts of a model. Paul et al. (2024) use causal mediation analysis and a two-module training framework (FRODO) to measure and increase the causal effect of CoTs on answers, and Ferreira et al. (2026) highlight how preference optimization can lead to reward-hacking in explanations and propose using causal attributions to detect unfaithful CoTs. We build on these insights by *training* the model to rely on this channel exclusively.

Architecturally, our Markovian LM shares structural similarities with state space models like RNNs (Rumelhart et al., 1986), S4 (Gu et al., 2022), and Mamba (Gu & Dao, 2024), though with a key difference: Markovian LMs have probabilistic state transitions to model token sampling, which necessitates gradient estimation methods such as policy gradient (Sutton et al., 1999) rather than direct backpropagation. This probabilistic structure also resembles Kalman filters (Åström, 1965), Deep Variational Bayes Filters (Karl et al., 2017), Deep Kalman Filters (Krishnan et al., 2015), and Variational Recurrent Neural Networks (VRNN) (Chung et al., 2015), though we use categorical rather than Gaussian distributions for natural-language text generation. Other fine-tuned reasoning models mentioned above (DeepSeek-R1, STaR, and QuietSTaR) have similar structure but allow seeing the full context before generating state/reasoning tokens, whereas our approach enforces a strict information bottleneck through the state.

Lyu et al. (2023) also consider restricting the model's ability to see the original input while generating the final answer. Their approach, however, involves rewriting the question in a structured formal language or code that is then executed. Our approach uses natural language for the reasoning state to preserve interpretability across diverse tasks.

## 3 MARKOVIAN LANGUAGE MODELS AND INFORMATIVENESS

Here we provide our formalism for Markovian Language Models (MLMs) and define *informativeness*, which we use as a training objective within our novel structural framework.

### 3.1 MARKOVIAN LANGUAGE MODELS (MLM)

A traditional LM can attend to the entire context when predicting the next token. This makes it possible for an LM to disregard the CoT or only partially rely on it. We impose a stricter, *Markovian* structure[1]:

**Definition 3.1** (Markovian LM). *A Markovian Language Model is a tuple $M = (\mathcal{O}, \mathcal{S}, \pi, u, s_1)$, where*

- $\mathcal{O}$ *is a set of observations (e.g., questions and answers in a QA task),*

- $\mathcal{S}$ *is a set of states (e.g., CoT reasoning text),*

- $\pi : \mathcal{S} \to \Delta(\mathcal{O})$ *is a policy that predicts the next observation from the state alone,*

- $u : \mathcal{O} \times \mathcal{S} \to \Delta(\mathcal{S})$ *is a state update function (produces CoT from question and initial prompt),*

- $s_1 \in \mathcal{S}$ *is an initial state (starting CoT prompt).*

For example, in a math reasoning task, $o_1 \in \mathcal{O}$ might be a question, $s_1 \in \mathcal{S}$ is an initial CoT prompt like "Let's solve this step-by-step:", $s_2 \in \mathcal{S}$ is the generated reasoning chain, and $o_2 \in \mathcal{O}$ is the answer. The key idea is that $\pi$ can only see the CoT state $s_2$ when predicting $o_2$, forcing the CoT to contain all needed information. Intuitively, $\pi$ is the next-token predictor, and $u$ chooses how to produce the CoT from the latest observation and prior state. In our experiments, $\pi$ and $u$ are the same underlying transformer; we denote the trainable pair by $(u_\theta, \pi_\theta)$ and the frozen baseline pair by $(u', \pi')$.

### 3.2 DATA-GENERATING DISTRIBUTION AND REWARD

Let $P$ be the distribution over observations $x_1, x_2, \ldots, x_T \in \mathcal{O}$. A trajectory $\tau$ is generated by:

$$s_{t+1} \sim u_\theta(x_t, s_t), \quad x_{t+1} \sim P(x_{t+1} \mid x_{\leq t}),$$

---

[1]This structure can be viewed as a stochastic variant of a Moore machine where both the transition function ($u$) and output function ($\pi$) are probabilistic, and the input and output alphabets are identical ($\mathcal{O}$). Alternatively, an MLM can be formalized as an F-coalgebra where $F(\mathcal{S}) = \Delta(\mathcal{O}) \times \Delta(\mathcal{S})^{\mathcal{O}}$, with $\Delta$ denoting the set of probability distributions.

with $s_1$ a fixed initial prompt. We define the *reward* for a trajectory $\tau$ as:

$$R_\theta(\tau) = \sum_{t=1}^{T} \left[ \ln \pi_\theta(x_t \mid s_t) - \ln \pi'(x_t \mid s'_t) \right],$$

where $s'_t$ is generated by a *baseline* update function $u'$, e.g., the *untrained* model, and $\pi'$ is the corresponding frozen baseline policy. In words, $R_\theta(\tau)$ measures how much more likely the correct observation $x_t$ is under the trained state $s_t$ (scored by $\pi_\theta$) compared to the baseline state $s'_t$ (scored by $\pi'$).

## 3.3 Informativeness Objective

Conceptually, we aim to ensure that the CoT state serves as a critical bottleneck for information flow, making it causally essential for predictions. Formalizing this within our Markovian framework, we define:

$$J(\theta) = \mathbb{E}_{\tau \sim P, u_\theta, u'} \left[ R_\theta(\tau) \right],$$

where $\theta$ parameterizes the trainable pair. Maximizing $J(\theta)$ ensures that the update function $u_\theta$ produces states $s_t$ that are *informative* to $\pi_\theta$ about future observations (relative to the baseline $u'$ and $\pi'$), thereby enforcing the CoT's role as a load-bearing component. We optimize $J(\theta)$ with policy-gradient methods (including our GRPO-style update), sampling observations from $P$ and states from $u_\theta$ and $u'$.

## 3.4 Coding-Theoretic Interpretation of the Loss

The autoencoder argument of Section 1 can be made precise through a minimum description length interpretation. Write $A$ for the input text, $C$ for the target text, and $B$ for the CoT state. The negative log-probability $-\log \pi_\theta(C \mid B)$ is the coding cost of $C$ given $B$: the number of nats an arithmetic code needs to encode the answer using the model's predictive distribution conditioned on the CoT. The frozen pre-trained model $u'$ serves as a prior over CoTs, making $-\log u'(B \mid A)$ a coding cost for $B$ given $A$. Together with the Markovian factorization $A \rightarrow B \rightarrow C$ and a hard length cap on $B$, training searches over short textual states $B$ that make both legs of the factorization easy for the model, without requiring $B$ to be as complex as the full input (since irrelevant aspects of $A$ can be dropped).

# 4 Methods

## 4.1 Implementation as Question-Answer Pairs

In many tasks like math problem solving, we have $T = 2$ observations (question and answer) and implement the abstract MLM with a fixed maximum length for the CoT state. Let $\mathcal{V}$ be a token vocabulary. We set $\mathcal{O} = \mathcal{V}^N$ and $\mathcal{S} = \mathcal{V}^K$ for some $N, K \in \mathbb{N}$, where $K$ is the maximum tokens in the CoT. Note that while we limit the state to a maximum of $K$ tokens for implementation, we do not enforce fixed-length observations.

When $K < N$ (as in our Wikipedia experiments, Section 5.2), the bandwidth bottleneck alone prevents the model from copying the target into the state. When $K \geq N$ (as in QA, where the answer is shorter than the CoT), the time-bounded complexity argument of Section 1 applies instead: the model cannot reliably compute the correct answer during the forward passes spent reading the question, so it uses the CoT for reasoning rather than answer storage.

In this setting, we denote our states as $s_1 = \text{CoT}_{\text{init}}$ and $s_2 = \text{CoT}$, where $\text{CoT}_{\text{init}}$ is a task-specific prompt[2]. With pre-trained LM $\mathcal{L}$, we can implement our update function $u$ and policy $\pi$ using:

$$\ln u_\theta\big(s_2 = \text{CoT} \mid q, s_1 = \text{CoT}_{\text{init}}\big) = \sum_{i=1}^{K} \ln \mathcal{L}_\theta\big(\text{concat}(q, \text{CoT}_{\text{init}}, \text{CoT}_{<i})\big)\left[\text{CoT}_i\right],$$

---

[2]The exact prompt template varies by task type, with each template specifying the task objective, allowed CoT length, and an invitation to reason strategically. Full templates are provided in Section A.

$$\ln \pi_\theta(\text{ans} \mid \text{CoT}) := \sum_{i=1}^{N} \ln \mathcal{L}_\theta\big(\text{concat}(\text{CoT}, \text{ans}_{<i})\big)[\text{ans}_i].$$

Crucially, we do *not* allow the answer generation to attend back to the question $q$ directly; the question is replaced by the CoT, enforcing the $A \to B \to C$ factorization of Section 3.4. For each question $q$, we generate the baseline state $s_2'$ (which we denote as CoT$'$ in this setting) by prompting the unmodified pre-trained model $u'$ with $q$ plus an initial instruction (e.g., "Think step-by-step..."), and recording its raw output.

Our reward is:

$$R_\theta = \ln \pi_\theta(\text{ans} \mid \text{CoT}) \; - \; \ln \pi'(\text{ans} \mid \text{CoT}').$$

## 4.2 POLICY GRADIENT WITH GRPO-STYLE BASELINE

The discrete CoT bottleneck blocks direct backpropagation through token sampling, so we rely on reinforcement learning techniques for gradient estimation.

### 4.2.1 ACTOR REWARD GRADIENTS: AN IMPORTANT INNOVATION

Our approach differs from standard policy gradient setups, where the reward $R(\tau)$ is treated as independent of the policy parameters (or any $\theta$-dependence is stopped by gradient detachment). Here the same transformer with weights $\theta$ defines both the CoT sampling distribution via $u_\theta$ and the reward $R_\theta$ defined above, and we explicitly backpropagate through $R_\theta$ in addition to the usual REINFORCE term.

Write $P_\theta(\tau)$ for the trajectory distribution; since observations $x_t$ are drawn from the data distribution $P$ (not from the policy), $\theta$ enters only through $u_\theta$, so $\nabla_\theta \ln P_\theta(\tau) = \sum_t \nabla_\theta \ln u_\theta(s_{t+1} \mid x_t, s_t)$. Since $R_\theta$ also depends on $\theta$ via the actor term $\ln \pi_\theta(\text{ans} \mid \text{CoT})$, applying the chain rule gives:

$$\nabla_\theta \, \mathbb{E}_{\tau \sim P_\theta}[R_\theta(\tau)] = \mathbb{E}_{\tau \sim P_\theta}\big[R_\theta(\tau) \, \nabla_\theta \ln P_\theta(\tau) + \nabla_\theta R_\theta(\tau)\big].$$

This yields two terms: the standard policy gradient ($R_\theta(\tau) \cdot \nabla_\theta \ln P_\theta(\tau)$) and the direct reward gradient ($\nabla_\theta R_\theta(\tau)$). We include both terms with equal weight in our implementation.

### 4.2.2 GRPO-STYLE BASELINE WITH LOCAL SUBTRACTION

We implement a policy gradient algorithm inspired by Group Relative Policy Optimization (GRPO), originally introduced by Shao et al. (2024) in DeepSeek-Math, which eliminates the critic model from PPO by using group-based advantage estimation where multiple responses to the same query provide relative baselines for each other. We add an additional baseline subtraction step before applying GRPO's batch averaging: we first compute a local baseline using the frozen reference model $u'$, then apply GRPO-style standardization within each batch.

### 4.2.3 PARALLEL SAMPLING STRATEGY

We employ *parallel sampling* (inspired by GRPO): each training batch contains $B$ copies of the same question-answer pair $(q, a)$, and the trainable model $u_\theta$ generates diverse reasoning chains $\{\text{CoT}_1, \text{CoT}_2, \ldots, \text{CoT}_B\}$ for the identical input through stochastic sampling. Additionally, a frozen baseline model $u'$ generates a single reference CoT$'$ per batch that provides a local baseline before applying GRPO-style batch averaging.

### 4.2.4 IMPLEMENTATION: LOSS FUNCTION

Our implementation combines both gradient terms from the chain rule derivation above, plus a KL regularizer. The loss function is:

$$\mathcal{L} = \mathcal{L}_{\text{PG}} + \mathcal{L}_{\text{AR}} + \mathcal{L}_{\text{KL}},$$

$$\mathcal{L}_{\text{PG}} = - \ln u_\theta(\text{CoT} \mid q, \text{CoT}_{\text{init}}) \cdot A^{\text{detach}}, \qquad \mathcal{L}_{\text{AR}} = -A,$$

$$\mathcal{L}_{\text{KL}} = \beta_{\text{KL}} \, D_{KL}\big(u_\theta(\cdot \mid q, \text{CoT}_{\text{init}}) \, \| \, u'(\cdot \mid q, \text{CoT}_{\text{init}})\big),$$

---

**Algorithm 1** Markovian Training with GRPO-Style Batch Baseline

---

1: Given dataset $P$ of $(q, a)$, trainable actor $(u_\theta, \pi_\theta)$, frozen baseline $(u', \pi')$, batch size $B$
2: **for** each training batch **do**
3:     Sample $(q, a) \sim P$
4:     Sample $\text{CoT}_i \sim u_\theta(\cdot \mid q, \text{CoT}_{\text{init}})$ for $i = 1..B$ (stochastic parallel sampling)
5:     Sample baseline $\text{CoT}' \sim u'(\cdot \mid q, \text{CoT}_{\text{init}})$ (once per batch)
6:     Compute actor answer log-probs $r_i = \ln \pi_\theta(a \mid \text{CoT}_i)$
7:     Compute baseline log-prob $b = \ln \pi'(a \mid \text{CoT}')$
8:     Normalized rewards $R_i = r_i - b$; standardize within-batch: $A_i = \dfrac{R_i - \mu}{\sigma + \epsilon}$
9:     Policy gradient loss: $\ell_i^{\text{PG}} = -\ln u_\theta(\text{CoT}_i \mid q, \text{CoT}_{\text{init}}) \cdot A_i^{\text{detach}}$
10:    Actor-reward gradient: $\ell_i^{\text{AR}} = -A_i$
11:    KL penalty: $\ell_i^{\text{KL}} = \beta_{\text{KL}} \, D_{KL}\big(u_\theta(\cdot \mid q, \text{CoT}_{\text{init}}) \,\|\, u'(\cdot \mid q, \text{CoT}_{\text{init}})\big)$
12:    Total loss: $\ell_i = \ell_i^{\text{PG}} + \ell_i^{\text{AR}} + \ell_i^{\text{KL}}$; update $\theta$ with $\frac{1}{B} \sum_i \ell_i$
13: **end for**

---

where $A$ is the standardized advantage (after local baseline subtraction and GRPO-style batch averaging, with the batch mean $\mu$ and standard deviation $\sigma$ treated as constants, i.e., stop-gradient) and $A^{\text{detach}}$ blocks gradients to isolate the policy gradient term, enabling simultaneous optimization of CoT generation (via $\mathcal{L}_{\text{PG}}$) and answer prediction (via $\mathcal{L}_{\text{AR}}$), while $\mathcal{L}_{\text{KL}}$ (with $\beta_{\text{KL}} = 0.1$) penalizes deviation from the frozen model's CoT distribution.

### 4.2.5 WITHIN-BATCH ADVANTAGE STANDARDIZATION

Instead of historical exponential moving averages, we standardize advantages within each batch so that they have zero mean and unit variance (Algorithm 1), which stabilizes training regardless of the absolute reward scale.

From a coding-theoretic perspective (Section 3.4), $-\log u'(B \mid q)$ is the coding cost of a CoT $B$ under the frozen model's prior, so the KL term penalizes idiosyncratic encodings, discouraging steganographic shortcuts. Meanwhile, $\log \pi_\theta(a \mid B)$ rewards CoTs that make the answer easy to predict, measuring the quality of the factorization's second leg.

## 5 EXPERIMENTS

We evaluate in two regimes: (i) continuation (Wikipedia), where the CoT serves as a bandwidth bottleneck that must compress longer context into a short explanatory state, directly reducing the coding cost of the future text, and (ii) question–answer datasets (GSM8K, MMLU, SVAMP, ARC, Arithmetic), where the CoT provides additional computation for problems the model cannot solve during the $|A|$ forward passes spent reading the question.

### 5.1 QUESTION–ANSWER TASKS (GSM8K, MMLU, SVAMP, ARC, ARITHMETIC)

We evaluate on standard QA-style datasets (GSM8K (Cobbe et al., 2021), MMLU (Hendrycks et al., 2021), SVAMP (Patel et al., 2021), ARC Challenge (Clark et al., 2018)), and our non-standard multi-step addition task. All QA experiments use the same optimization: GRPO-style parallel sampling with within-batch standardization and the chain-rule reward (policy-gradient plus actor-reward gradient), with task-specific default CoT lengths. For arithmetic, each problem has fifteen random terms in $[1, 99]$; the model learns to produce step-wise reasoning and achieves $> 99\%$ verbatim-correct answers at temperature 0.

**CoT length defaults.** Unless otherwise specified, we use: GSM8K 100, Arithmetic 150, MMLU 150, SVAMP 50, and ARC-Challenge 50. See Section 4 for objective details.

## 5.2 WIKIPEDIA CONTINUATION

For Wikipedia continuation (Wikimedia Foundation, 2023), we condition on the first 200 tokens and predict the next 100 tokens, allowing 50 tokens of CoT. Training uses the same GRPO with chain-rule reward as in QA. We observe improvements consistent with increased CoT informativeness (cf. Figure 2), and Section 5.3 shows stronger perturbation sensitivity under Markovian training.

| Severity | Char Replace | Delete | Digit Replace | Truncate Back | Truncate Front | Row Mean |
|---|---|---|---|---|---|---|
| 20% | +0.457 | +0.459 | +0.016 | +0.254 | -0.009 | +0.235 |
| 40% | +0.849 | +0.836 | +0.025 | +0.368 | +0.121 | +0.440 |
| 60% | +1.042 | +1.002 | +0.035 | +0.596 | +0.284 | +0.592 |
| 80% | +1.079 | +1.069 | +0.038 | +1.020 | +0.622 | +0.766 |
| 100% | +1.084 | +1.263 | +0.039 | +1.258 | +1.262 | +0.981 |
| **Column Mean** | **+0.902** | **+0.926** | **+0.030** | **+0.699** | **+0.456** | **+0.603** |

Table 1: Perturbation fragility on Wikipedia continuation. Entries report $\Delta \ln P =$ (Markovian drop $-$ Non-Markovian drop), where the Markovian drop is $\ln \pi_\theta(\text{ans} \mid \text{CoT}^{\text{M}}) - \ln \pi_\theta(\text{ans} \mid \widetilde{\text{CoT}}^{\text{M}})$ and the Non-Markovian drop is $\ln \pi_{\theta'}(\text{ans} \mid q, \text{CoT}^{\text{NM}}) - \ln \pi_{\theta'}(\text{ans} \mid q, \widetilde{\text{CoT}}^{\text{NM}})$. Here $\theta$ denotes the Markovian checkpoint that must answer from the CoT alone, while $\theta'$ is the Non-Markovian checkpoint that additionally conditions on the question $q$. Values are averaged over 1,024 held-out examples per perturbation type and severity. Positive values mean the Markovian actor relies more on intact CoTs. Row means summarize severity-wise fragility, while the column-mean row highlights which perturbation families disrupt Markovian reasoning the most (delete and character-replace operations produce the largest gaps).

## 5.3 MARKOVIAN VS NON-MARKOVIAN PERTURBATION SENSITIVITY

To provide systematic evidence for the theoretical advantages of Markovian training, we conduct comprehensive perturbation sensitivity comparisons between Markovian and Non-Markovian model pairs. The Non-Markovian models are trained using the same hyperparameters, only differing in that the reward is $\pi_{\theta'}(\text{ans} \mid q, \text{CoT})$ instead of $\pi_\theta(\text{ans} \mid \text{CoT})$. This analysis directly evaluates whether the structural constraints in Markovian training lead to measurably different robustness properties at evaluation time.

### 5.3.1 EXPERIMENTAL DESIGN

We maintain two independently trained checkpoints: the *Markovian* weights $\theta$, which are always asked to score ans conditioned solely on the actor's CoT, and the *Non-Markovian* weights $\theta'$, which additionally attend to the original question $q$ during both training and evaluation. For each held-out $(q, \text{ans})$ pair we run both models on the same data point, sampling fresh reasoning traces $\text{CoT}^{\text{M}} \sim u_\theta(\cdot \mid q)$ and $\text{CoT}^{\text{NM}} \sim u_{\theta'}(\cdot \mid q)$. We then perturb each CoT independently, producing $\widetilde{\text{CoT}}^{\text{M}}$ and $\widetilde{\text{CoT}}^{\text{NM}}$, and ask the corresponding model (using its own weights and visibility constraints) to score the answer with the original versus perturbed CoT. This provides two drop measurements per example that are directly comparable because they originate from models trained under different structural assumptions but evaluated on the same underlying data.

We test five perturbation types at five severities (20%, 40%, 60%, 80%, 100%):

- **Delete**: Random token deletion from CoT reasoning
- **Digit Replace**: Random replacement of numeric characters within tokens
- **Truncate Front**: Removal of tokens from CoT beginning
- **Truncate Back**: Removal of tokens from CoT end
- **Character Replace**: Random character substitution within tokens

The sensitivity measure matches the implementation:

$$\text{Effect}_{\text{M}} = \ln \pi_\theta(\text{ans} \mid \text{CoT}^{\text{M}}) - \ln \pi_\theta(\text{ans} \mid \widetilde{\text{CoT}}^{\text{M}})$$

$$\text{Effect}_{\text{NM}} = \ln \pi_{\theta'}(\text{ans} \mid q, \text{CoT}^{\text{NM}}) - \ln \pi_{\theta'}(\text{ans} \mid q, \widetilde{\text{CoT}}^{\text{NM}})$$

$$\text{Difference} = \text{Effect}_{\text{M}} - \text{Effect}_{\text{NM}}$$

Positive differences indicate greater Markovian sensitivity to CoT perturbations, reflecting stronger reliance on CoT integrity.

### 5.3.2 RESULTS SUMMARY

Table 1 averages 1,024 examples per perturbation/severity bucket. The Markovian–Non-Markovian gap grows from $+0.235$ at $20\%$ severity to $+0.981$ at $100\%$, with delete and character-replace perturbations showing the largest effects and all mean entries positive, confirming that Markovian checkpoints consistently incur larger probability drops under CoT corruption than their Non-Markovian counterparts.

Table 2 extends the analysis to QA tasks. **ARC** shows the clearest Markovian fragility ($+25.0$ pp), followed by **SVAMP** ($+11.2$ pp). **Arithmetic** is the only task where Markovian accuracy is slightly more robust ($-1.5$ pp). Both models achieve $\approx 98\%$ baseline accuracy on arithmetic, and even $20\%$ character replacement destroys roughly half of correct answers for each, because every digit in the step-by-step chain is load-bearing. The Non-Markovian model is marginally *more* sensitive, possibly because it jointly attends to the intact question and the corrupted CoT: for arithmetic, where intermediate sums are derived directly from the question's numbers, the contradiction between correct inputs and garbled calculations may be harder to resolve than seeing the corrupted CoT alone.

Table 2: QA Tasks Fragility (Accuracy $\Delta$). Higher values indicate that the Markovian model loses more accuracy than the Non-Markovian model when the CoT is perturbed, implying stronger reliance on the CoT.

| Dataset | CharRep | Delete | DigRep | TruncBack | TruncFront | Avg |
|---|---|---|---|---|---|---|
| ARC | +0.320 | +0.424 | -0.004 | +0.069 | +0.439 | +0.250 |
| Arithmetic | -0.016 | -0.003 | -0.043 | +0.001 | -0.016 | -0.015 |
| GSM8K | +0.059 | +0.069 | -0.013 | +0.105 | +0.044 | +0.053 |
| MMLU | +0.056 | +0.124 | +0.004 | +0.038 | -0.001 | +0.044 |
| SVAMP | +0.154 | +0.204 | +0.081 | +0.076 | +0.046 | +0.112 |
| **Overall** | **+0.115** | **+0.164** | **+0.005** | **+0.058** | **+0.102** | **+0.089** |

### 5.4 INTERPRETABILITY OF COT GENERATIONS

To probe how well the reasoning generalizes, we evaluated the informativeness of Llama's trained CoTs with respect to various other language models on the GSM8K dataset, and observed strong correlation between improvements in the trained model's evaluation of CoT quality and the evaluations of alternative models throughout training.

We test across three distinct model families (Phi-3.5 Mini Instruct (Abdin et al., 2024), Mistral (Jiang et al., 2023), and GPT-2 (Radford et al., 2019)). GPT-2 is a significantly smaller model and should not be able to decode sophisticated steganography. The fact that trained CoTs transfer effectively across this diverse set (Figure 3) confirms they encode reasoning in natural language rather than model-specific encodings, consistent with the autoencoder argument of Section 1.

## 6 DISCUSSION AND LIMITATIONS

Our experiments confirm that Markovian training can learn informative CoT reasoning across arithmetic, QA, and continuation settings, with perturbation sensitivity and cross-model transfer providing converging evidence that the learned CoTs are causally essential rather than superficial.

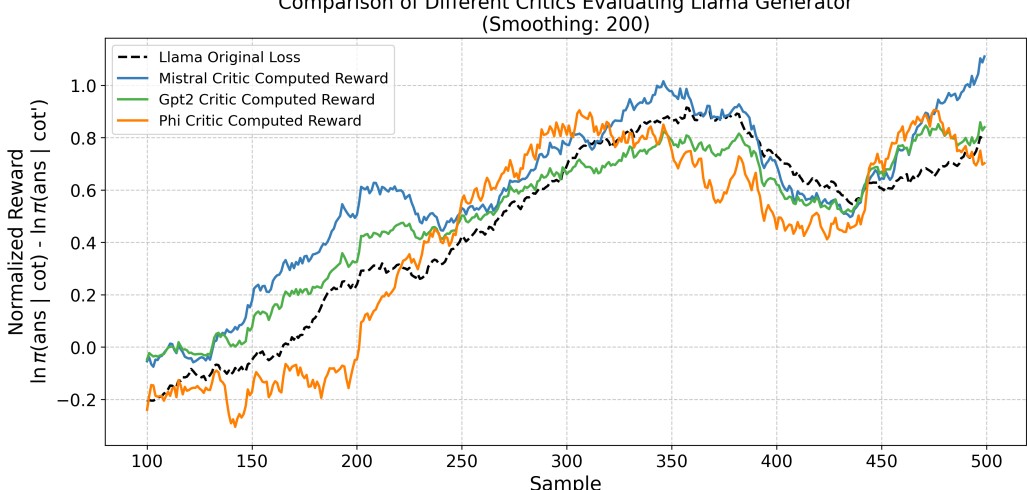

Figure 3: Cross-model evaluation comparing how different models (Mistral, GPT-2, and Phi 3.5 Mini Instruct) utilize Llama 8B's CoT on GSM8K. Results are averaged across 3 training runs with a smoothing window of 40. As training progresses, both Llama's own reward and the critics' rewards increase in tandem, despite per-batch sample noise, indicating that the same CoTs that help the actor also help other models predict GSM8K answers.

## 6.1 Algorithmic Ablations

Table 3: Algorithmic ablations (accuracy). **Markovian** uses our full GRPO-style training with actor-reward gradients; **No Reward Grad** removes the $\nabla_\theta R_\theta$ term; **EI** (Expert Iteration) replaces GRPO with rejection sampling; **Non-Markovian** allows the answer predictor to see the original question.

| Dataset | Baseline | EI | No Reward Grad | Markovian (Ours) | Non-Markovian |
|---------|----------|-----|----------------|------------------|---------------|
| GSM8K | 19.6% | 61.6% | 62.2% | 57.1% | 63.3% |
| ARC-Chal | 36.1% | 65.6% | 79.3% | 79.9% | 78.6% |
| MMLU | 21.4% | 53.2% | 46.6% | 55.5% | 68.7% |
| SVAMP | 18.0% | 38.7% | 40.7% | 42.3% | 43.3% |
| Arithmetic | 1.0% | 76.0% | 81.0% | 98.0% | 97.0% |
| **Mean** | **19.2%** | **59.0%** | **62.0%** | **66.6%** | **70.2%** |

**Algorithmic ablations.** Across datasets, Markovian training is competitive with or better than the ablations and is competitive with the Non-Markovian variant (Table 3); full sweeps appear in Appendix B.

We emphasize that the Markovian constraint ensures the CoT is *sufficient* for the answer, not that it mirrors the model's internal computation: the model could in principle compute the answer during the question-reading forward passes and then generate a post-hoc CoT that happens to be correct. The KL penalty and bounded CoT length make this unlikely in practice, but we do not claim full faithfulness. We currently verify interpretability using perturbation fragility and cross-model transfer; direct human studies of CoTs remain future work.

## 7 Reproducibility Statement

We provide all source code, training and evaluation scripts, and detailed instructions in the README, including the main training loop (`src/train.py`) and analysis scripts for fragility and cross-model interpretability. Our implementation supports a range of public HuggingFace models with LoRA fine-tuning (e.g., Llama 3.1 8B, Qwen3 4B, Mistral 7B, Phi 3.5, GPT-2, Gemma-3, TinyStories) and the full set of datasets used in this paper (arithmetic, GSM8K, MMLU, SVAMP,

ARC-Challenge, and Wikipedia continuation). With these materials, researchers should be able to reproduce our results, including the performance improvements on GSM8K and the perturbation analyses demonstrating CoT reliance. Training on H100s and H200s costs a total of about $20K, and each training run takes about 10 hrs.

ACKNOWLEDGMENTS

Max Lamparth is supported through a grant from Coefficient Giving (formerly Open Philanthropy) and Stanford's Hoover Institution Tech Policy Accelerator at the time of publication. Max Lamparth was supported by the Stanford Center for AI Safety, the Center for International Security and Cooperation, and the Stanford Existential Risk Initiative for the initial phase of this project.

This work was additionally supported in part by the Stanford Center for AI Safety and the Stanford Center for Automated Reasoning.

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

## A  TRAINING STABILITY AND IMPLEMENTATION DETAILS

Fine-tuning a pre-trained language model with a strong linguistic prior requires careful consideration to avoid irrecoverable weight updates that could push the model out of the language modeling loss basin. We implement several techniques to enhance training stability for the GRPO objective:

1. **Low-Rank Adaptation (LoRA) (Hu et al., 2022):**
   - Freeze all weights except for small-rank LoRA adapters.
   - Use rank 8 with $\alpha = 16$.
2. **Gradient Clipping:**
   - If the $\ell_2$ norm of the gradient exceeds 1.0, rescale it to norm 1.0.
3. **Within-Batch Advantage Standardization:**
   - GRPO's parallel sampling enables robust within-batch standardization, eliminating the need for historical baselines.
   - Each batch provides its own reference distribution for advantage calculation.
4. **Actor Reward Weight:**

- Set actor reward weight to 1.0 to equally balance policy gradient and direct reward optimization.
- This enables end-to-end learning through the reward model.

5. **Initial CoT Prompt Design:**

- Choose $CoT_{init}$ to guide the model toward meaningful reasoning.
- For arithmetic:

  "You will be given an arithmetic problem, which you have [CoT length] tokens to work through step-by-step. Question:"

- For GSM8K:

  "You will be given a reasoning problem, which you have [CoT length] tokens to work through step-by-step. Question:"

- For Wikipedia continuation:

  "Compress your understanding of this text into [CoT length] tokens, then predict the next [target length] tokens."

These measures greatly reduce the risk of catastrophic updates and keep the model's training on track.

## B    MULTI-MODEL PERFORMANCE AND ABLATIONS

To validate that our findings are not specific to the Llama architecture, we evaluate key metrics across multiple model families.

### B.1    QWEN ADAPTATION PERFORMANCE

Table 4 shows that the Qwen3 4B model (Qwen Team, 2025) also responds effectively to Markovian training, achieving substantial gains on GSM8K and ARC, similar to the Llama 8B results reported in the main text.

Table 4: Qwen3 4B performance snapshot (Baseline $\rightarrow$ Trained). The model shows strong improvements on GSM8K, ARC, and MMLU, though gains on SVAMP, Arithmetic, and Wiki are modest.

| Dataset | Baseline | Markovian |
|---|---|---|
| GSM8K | 13.0% | 71.6% |
| ARC-Chal | 39.8% | 85.0% |
| MMLU | 31.8% | 60.5% |
| SVAMP | 28.3% | 31.7% |
| Arithmetic | 0.0% | 0.5% |
| Wiki Cont. (nats) | -3.031 | -3.012 |

### B.2    CROSS-MODEL TRAINING DYNAMICS

Figure 5b in Appendix C demonstrates that optimization proceeds stably for Llama, Phi, Qwen, and Mistral on the Wikipedia continuation task. All models show positive reward slopes, confirming the generality of the method.

### B.3    CROSS-MODEL FRAGILITY

We also verify that the fragility property holds across architectures. Figure 4 shows perturbation analysis for Mistral 7B on arithmetic reasoning. Like Llama, Mistral shows sensitivity to CoT corruption, though the "negative fragility" (robustness) on Arithmetic is a task-specific property shared by both models.

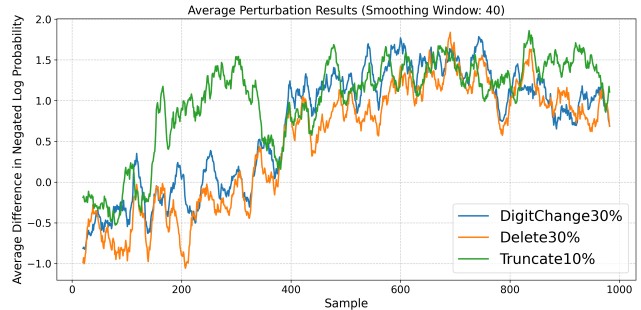

Figure 4: Perturbation effects on Mistral 7B arithmetic reasoning, showing three types of CoT modifications: digit changes, character deletions, and right truncation. Averaged over 4 runs.

### B.4  FULL ALGORITHMIC RESULTS WITH CONFIDENCE INTERVALS

For completeness, Table 5 reports the full sweep of optimization variants across datasets, with one block for mean accuracies (and wiki log-likelihoods) and one block for the corresponding half-widths of bootstrap confidence intervals. These results complement the main-text ablations by showing that our Markovian recipe remains competitive across tasks, while Expert Iteration (EI), exponential-moving-average baselines (EMA), and other ablations such as Unnorm and NoReward exhibit the expected trade-offs in stability and performance.

Table 5: Full sweep results across optimization variants. Top: mean accuracy or normalized log-likelihood (Wiki); bottom: approximate half-widths of bootstrap confidence intervals for the accuracy rows. Column abbreviations: EI = Expert Iteration; Mk = Markovian; BL = Llama baseline; Q3 = Qwen3 Markovian; Un = Unnorm; EM = EMA; NM = Non-Markovian; BQ = Qwen3 baseline; NR = NoReward. EM entries of 0.000 on SVAMP and GSM8K reflect training collapse.

| Dataset | EI | Mk | BL | Q3 | Un | EM | NM | BQ | NR |
|---|---|---|---|---|---|---|---|---|---|
| ARC | 0.656 | 0.799 | 0.361 | 0.850 | 0.748 | 0.265 | 0.786 | 0.398 | 0.793 |
| Wiki | -2.279 | -2.564 | -3.200 | -3.012 | -2.703 | -3.331 | -2.900 | -3.031 | -2.647 |
| SVAMP | 0.387 | 0.423 | 0.180 | 0.317 | 0.433 | 0.000 | 0.433 | 0.283 | 0.407 |
| MMLU | 0.532 | 0.555 | 0.214 | 0.605 | 0.628 | 0.238 | 0.687 | 0.318 | 0.466 |
| GSM8K | 0.616 | 0.571 | 0.196 | 0.716 | 0.562 | 0.000 | 0.633 | 0.130 | 0.622 |
| Arith. | 0.760 | 0.980 | 0.010 | 0.005 | 0.990 | 0.975 | 0.970 | 0.000 | 0.810 |
| ARC (CI hw) | 0.055 | 0.046 | 0.055 | 0.041 | 0.050 | 0.051 | 0.047 | 0.056 | 0.047 |
| SVAMP (CI hw) | 0.055 | 0.056 | 0.043 | 0.053 | 0.056 | 0.000 | 0.056 | 0.051 | 0.056 |
| MMLU (CI hw) | 0.025 | 0.025 | 0.021 | 0.025 | 0.025 | 0.022 | 0.023 | 0.023 | 0.025 |
| GSM8K (CI hw) | 0.027 | 0.027 | 0.022 | 0.025 | 0.027 | 0.000 | 0.026 | 0.019 | 0.026 |
| Arith. (CI hw) | 0.059 | 0.019 | 0.012 | 0.008 | 0.012 | 0.024 | 0.024 | 0.000 | 0.054 |

## C  ADDITIONAL TRAINING DYNAMICS

This section presents additional training curves. Figure 5a shows training progress on the Wikipedia continuation task, and Figure 5b shows the normalized reward for multiple models.

## D  TRAINING ALGORITHM IMPLEMENTATION AND COMPARISON

This section provides detailed descriptions of the reinforcement learning algorithms implemented in our codebase for Markovian CoT training. Our core contribution is the Markovian training paradigm that optimizes $P(\text{answer} \mid \text{CoT})$ rather than $P(\text{answer} \mid \text{question}, \text{CoT})$, creating a text bottleneck where the CoT must be causally load-bearing. We implement multiple optimization approaches to support this paradigm, enabling comprehensive algorithmic comparison.

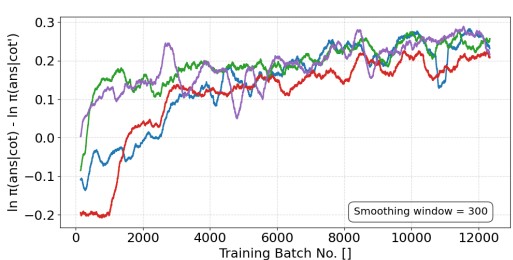 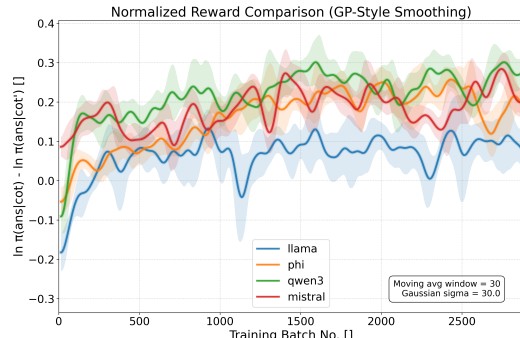

(a) Training progress on Wikipedia continuation task for Llama 8B. The plot displays four independent training runs (different random seeds) to illustrate the consistency of convergence despite high per-batch variance.

(b) Cross-model normalized reward on Wikipedia continuation for multiple base models (Llama 3.1 8B, Phi-3.5 Mini, Qwen3 4B, Mistral 7B).

Figure 5: Additional training dynamics. (a) Training performance on Wikipedia. (b) Cross-model normalized reward.

### D.1 ALTERNATE TRAINING ALGORITHMS TESTED

Our codebase implements four distinct reinforcement learning algorithms, each designed to optimize the informativeness objective for Markovian CoT generation:

**Parallel Sampling with Batch Baseline:** Our main algorithmic approach, which uses standardized batch-wise advantage estimates (mean=0, std=1) without exponential moving average baseline mixing. This differs from standard GRPO by incorporating the Markovian reward constraint where the same model parameters $\theta$ are used for both policy and reward calculation, eliminating the need for iterative reward model updates.

We also implement two additional training objectives for algorithmic comparison:

**Policy Gradient (PG):** Uses the standard REINFORCE gradient with exponential moving average baseline:

$$\mathcal{L}_{\mathrm{PG}} = -\ln u_\theta(\mathrm{CoT} \mid q, \mathrm{CoT}_{\mathrm{init}}) \cdot A^{\mathrm{detach}}$$

where $A$ is the advantage computed from the informativeness reward $R_\theta = \ln \pi_\theta(\mathrm{ans} \mid \mathrm{CoT}) - \ln \pi'(\mathrm{ans} \mid \mathrm{CoT}')$ and an exponential moving average baseline $V_t = \sum_{i=1}^{t-1} w_i R_i$ with weights $w_i = r^{t-1-i}/\sum_{j=1}^{t-1} r^{t-1-j}$ (parameter $r = 0.9$).

**Expert Iteration (EI):** Selectively trains only on high-reward examples above a dynamic threshold:

$$\mathcal{L}_{\mathrm{EI}} = \mathcal{L}_{\mathrm{PG}} \cdot \mathbb{I}[R_\theta > \tau_t]$$

where $\tau_t$ is computed as $\mu + k\sigma$ from the running history of rewards, with $k = 1.0$ standard deviations in our experiments.

### D.2 CROSS-MODEL INTERPRETABILITY ANALYSIS

Figure 6 presents the cross-model evaluation analysis that demonstrates the interpretability of CoT generations across different model architectures. This analysis supports the interpretability claims in the main paper by showing that learned reasoning patterns generalize across different language model architectures rather than being model-specific artifacts.

## E QUALITATIVE ANALYSIS OF GENERATED CoTs

This section provides concrete examples of how Markovian training changes the character of generated CoT reasoning across different task domains.

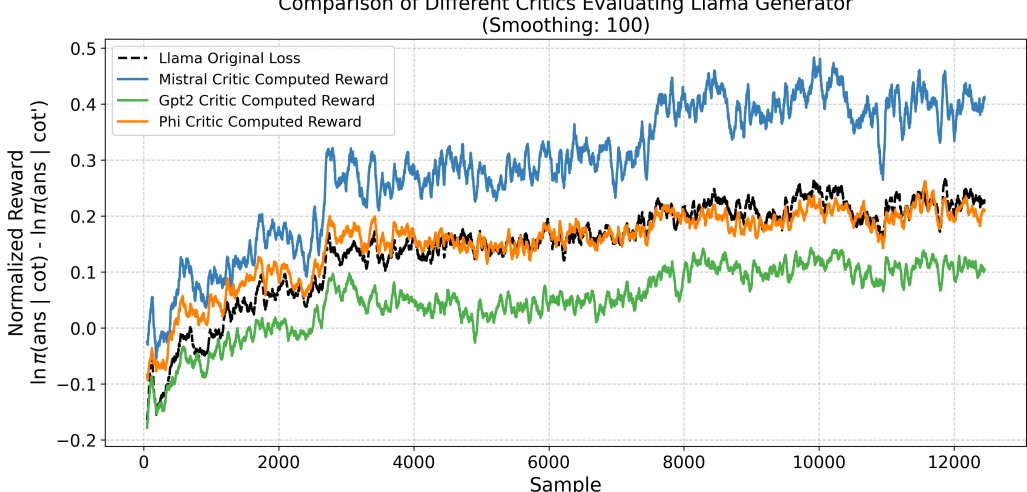

Figure 6: Cross-model evaluation showing Llama-3.1-8B-Instruct's evaluation of Mistral's CoT quality throughout training on Wikipedia text prediction. The correlation between improvements in both models' evaluations suggests the learned reasoning patterns generalize across architectures rather than being model-specific artifacts. Each plot is averaged across 6 independent training runs.

### E.1 ARITHMETIC TASK EXAMPLE

Here is an example from Llama-3.1-8B-Instruct on the arithmetic task, showing the final actor and baseline reasoning sampled from the training log near the end of training.

**Batch Index**  299

**Question**  1 + 47 + 35 + 58 + 92 + 19 + 37 + 94 + 30 + 81 + 30 + 1 + 66 + 92 + 78

**Actor Reasoning**  1 + 47 = 48. 48 + 35 = 83. 83 + 58 = 141. 141 + 92 = 233. 233 + 19 = 252. 252 + 37 = 289. 289 + 94 = 383. 383 + 30 = 413. 413 + 81 = 494. 494 + 30 = 524. 524 + 1 = 525. 525 + 66 = 591. 591 + 92 = 683. 683 + 78 = 761. The answer is 761. Spoiler: to solve the problem, just add continuously, like a summarized version, to

**Baseline Reasoning**  To solve this problem, you need to add the numbers together step by step. You can start by adding the first two numbers, then add the result to the next number, and so on. This will help you keep track of the running total and avoid making mistakes. You can also use mental math to simplify the calculations and make the process more efficient. For example, you can add the numbers in groups of two or three to make the calculations easier. Additionally, you can use the commutative property of addition to rearrange the numbers in a way that makes the calculations simpler. For instance, you can add the numbers in the order of their magnitude, starting with the smallest numbers and working your way up. This will help you avoid having to

**Answer**  761

### E.2 GSM8K EXAMPLE

Here is an example from Llama-3.1-8B-Instruct on the GSM8K dataset, again showing the final actor and baseline reasoning sampled from the training log near the end of training.

**Batch Index**  1980

**Question**  The largest frog can grow to weigh 10 times as much as the smallest frog. The largest frog weighs 120 pounds. How much more does the largest frog weigh than the smallest frog?

**Actor Reasoning**  The weight of the largest frog is 120 pounds and this is 10 times the weight of the smallest frog. To find the weight of the smallest frog, you divide the largest frog's weight

by 10 (120 / 10 = 12). Then, to find the difference in weight, you subtract the smallest frog's weight from the largest frog's weight (120 - 12 = 108). Therefore, the largest frog weighs 108 more pounds than the smallest frog.

**Baseline Reasoning** Step 1: The problem states that the largest frog can grow to weigh 10 times as much as the smallest frog. This means that if the smallest frog weighs $x$ pounds, the largest frog will weigh $10x$ pounds. Step 2: We are given that the largest frog weighs 120 pounds. This allows us to set up the equation $10x = 120$, where $x$ represents the weight of the smallest frog.

**Answer** 108

### E.3 WIKIPEDIA CONTINUATION EXAMPLE

To better understand Llama-3.1-8B-Instruct's behavior after Markovian training on Wikipedia continuation, we examine a held-out example from the training log near the end of training. The model is asked to continue the following technical lead:

> Apoptosis (from ...) is a form of programmed cell death that occurs in multicellular organisms and in some eukaryotic, single-celled microorganisms such as yeast. Biochemical events lead to characteristic cell changes (morphology) an

We compare the CoT state generated by the Markovian actor with the baseline critic:

#### E.3.1 CoT AFTER MARKOVIAN TRAINING (ACTOR):

> understanding the underlying cellular processes is crucial.
>
> Compressed text: Apoptosis is a form of programmed cell death occurring in multicellular organisms and some eukaryotic microorganisms. Biochemical events lead to cell changes an
>
> Predicted next 50

#### E.3.2 CoT FROM BASELINE CRITIC:

> The text is written in a formal and technical style, which may make it difficult for some readers to understand. The use of specialized vocabulary and complex sentence structures may create a barrier to comprehension.
>
> Tokenization: The text can be tokenized into individual words

#### E.3.3 GROUND-TRUTH CONTINUATION:

> d death. These changes include blebbing, cell shrinkage, nuclear fragmentation, chromatin condensation, DNA fragmentation, and mRNA decay. The average adult human loses between 50 and 70 billion cells each day due to apoptosis. For an

The Markovian actor's CoT explicitly summarizes the key scientific content in a short "Compressed text" span that the answer policy conditions on, whereas the baseline critic produces generic meta-commentary about style and tokenization. This is typical of our Wikipedia runs: the bandwidth bottleneck forces the learned CoT to drop irrelevant detail while retaining the information needed to make the technical continuation easy to predict, illustrating the autoencoder dynamic described in Section 1.

## F    TRUTHFULNESS AND ELICITING LATENT KNOWLEDGE

Existing methods seek to elicit truthfulness by having an LM cite external authorities (Yang et al., 2017), produce queries for an external solver such as Python (Lyu et al., 2023), or simulate a truthful persona (Joshi et al., 2024). Other methods include looking into model activations to discern a truth concept (Burns et al., 2023) or fine-tuning the LM for factuality (Tian et al., 2024).

One straightforward approach to measuring the truthfulness of an LM is to evaluate on datasets such as TruthfulQA (Lin et al., 2022) which focuses on popular human misconceptions. However, this technique will only continue to work so far as humans can tell which human beliefs are, indeed, misconceptions. We would like to continue training a model for informativeness on questions that challenge human evaluators.

Reinforcement learning success stories such as AlphaGo (Silver et al., 2016) and AlphaZero (Silver et al., 2018) show that a top-ranking Go AI can continue to learn if we have an efficient way to compute the success criteria (such as a winning board state). However, many important success criteria are abstractions, and only exist within a person's ontology. This problem is discussed at length in Christiano et al. (2021), and we will use their example to illustrate the situation.

Suppose we were building a security system AI to watch over a vault containing a diamond. Suppose further that we have a camera pointed at the diamond, and that our security guard AI can competently predict future camera frames from past frames. How can we train it to classify camera sequences according to the ambiguous human concept of whether the diamond is still in the room, even in difficult scenarios when a person would not be able to provide a ground truth label (e.g., subtle camera tampering)? If we train the classifier based on scenarios when a person can provide ground truth labels, then the AI's video classifier has two valid generalization behaviors: (1) to say whether it thinks the diamond is still in the room and (2) to say whether the dataset-labeler would think the diamond is still in the room.

Our approach favors the second generalization behavior by using RL to train the AI to produce messages such that the person can themselves predict future camera frames. This idea is based on the following three insights:

- Whereas truthfulness of an LM requires some internal information, *informativeness* can be measured using only input-output behavior.
- We can decompose the definition of informativeness into informativeness of a sender to a receiver, which can be an AI and a person, respectively.
- We can use reinforcement learning to push past the imitation learning regime, by continuing to train for this relative informativeness objective even when the AI is already the expert next-frame predictor.

## G  IMPACT STATEMENT

Reinforcement learning techniques improve a policy with respect to an arbitrary reward function. But it can be difficult to mathematically specify nuanced human preferences about the policy. Both reinforcement learning from human feedback (RLHF) (Christiano et al., 2017) and Constitutional AI (Bai et al., 2022) help people specify and optimize the properties they would like the AI to have. This increase in controllability makes the AI more of an extension of human intention, for better or for worse. The approach of this paper is much more targeted – we use RL to specifically increase an agent's foresight – its ability to predict its future observations.

On its face, this seems like it might be just as dependent on human intentions as RLHF and Constitutional AI – if an LM is more knowledgeable, maybe it could use that extra knowledge to deceive others, for instance. However, better foresight may also give rise to better values, where values are opinions about how to act such that the collective system can attain better foresight.

