# OpenReview forum: "Markovian Transformers for Informative Language Modeling"
_ICLR.cc/2026/Conference — ICLR 2026 Poster_

### Official Review · Reviewer_K7Fw · 2025-10-31

**Soundness:** 2
**Presentation:** 2
**Contribution:** 2
**Rating:** 4
**Confidence:** 3

**Summary:**

The paper proposes Markovian Language Model framework. By training the system with GRPO-style policy gradient algorithm, the model yields large gains on QA tasks.

**Strengths:**

The proposed markovian transformers is interesting and I recognize its novelty. The paper did abundant experiments on various datasets, which makes the paper sound.

**Weaknesses:**

W1: Though you mentioned what is the baseline in your manuscript, but I didn't understand what your baseline is quite well. Also, I want to look at the comparison between yours method/architecture between the model/training method we are now using today. For example, I didn't see the comparison to LLM post-trained on CoT by teacher-forcing and RL methods. It is hard for me to judge its significance.

**Questions:**

Q1: In figure 1, while $o_1$ stands for question and $s$ stands for CoT, it is hard for me to understand what is the $o_i$ on the right. How do you get the $o_i$? If it is sampled according to $s_i$, then which part corresponds to the question?

Q2: In figure 2, can you explain why it actually performs worse on MMLU?

---

> ### Author Response · Authors · 2025-11-25
>
> > W1: Though you mentioned what is the baseline in your manuscript, but I didn't understand what your baseline is quite well.
>
> Our main baseline is a Non-Markovian RL variant (GRPO-style without the Markovian bottleneck), which we have now added to Table 2. The “Baseline” column in that table corresponds to the frozen instruction-tuned model (standard SFT without RL).
>
> > Also, I want to look at the comparison between yours method/architecture between the model/training method we are now using today.
>
> As for other post-training methods, we include Expert Iteration (EI) and additional ablations in Table 2. EI is close in spirit to STaR-style CoT distillation (keep only high-reward CoTs and train on those), so together with the SFT baseline and the Non-Markovian GRPO variant this provides the requested comparisons.
>
> > Q1: In figure 1, while o_1 stands for question and stands for CoT, it is hard for me to understand what o_i is the on the right. How do you get the o_i? If it is sampled according to s_i, then which part corresponds to the question?
>
> On the right-hand side, $o_1,o_2,\dots$ denote a generic sequence of observations drawn from the data distribution. In the QA setting we instantiate this with $T=2$: $o_1$ is the question and $o_2$ is the answer. More generally, the data-generating process produces each $o_i$ given the previous observations $o_{<i}$; in our concrete QA case the question is always $o_1$ and the answer is $o_2$.
>
> > Q2: In figure 2, can you explain why it actually performs worse on MMLU?
>
> Since this is RL, individual training steps are slower, and in the original submission we did not get through a full epoch of the 99,842 MMLU points. In the revision we extended training (more datapoints and steps), which substantially improved the MMLU score; the updated figure now shows a significantly positive effect on MMLU.

---

### Official Review · Reviewer_V7a8 · 2025-11-01

**Soundness:** 2
**Presentation:** 3
**Contribution:** 3
**Rating:** 6
**Confidence:** 2

**Summary:**

This paper tries to improve the problem of chain of thought unfaithfulness, specifically when chain of thought isn't actually necessary for the model to reach its conclusions. The authors propose a Markovian framework wherein the model is blocked from attending to the original question during answer generation This forces all task-relevant information to be funneled through the chain of thought. To do this training, they use a GRPO-style algorithm and introduce further improvements like parallel sampling. After training, the models perform better on math reasoning benchmarks. The authors also use perturbation analyses on a Wikipedia text completion task to show that the chains of thought produced by their methods are more causally dependent than those produced by standard approaches.

**Strengths:**

- The problem (chain-of-thought unfaithfulness) is of broad interest to the field
- The method design seems novel and creative. It also seemed tricky to implement and the authors developed training strategies to make it work.

**Weaknesses:**

- Although the authors state that the approach improves performance on benchmarks, I don't think they compared to a baseline-- eg, what if you just did the training but didn't have the attention to the original question blocked?
- It would also be nice to have some qualitative discussions of the chains of thought that result from this training procedure, whether through example transcripts or through some grading of readability.

**Questions:**

- I'm not sure how to interpret figure 3. I'm assuming the x-axis ('sample') is number of tokens sample. Are there some examples of what these chain of thought transcripts look like? Are there parts of the transcript where the llama model has stated the answer?
- I also would like more clarification to interpret the perturbation analyses. Is this analysis testing how well the model adheres to the exact continuation of the wikipedia text? If so, why not do the same perturbation analyses on the previous QA tasks?

---

> ### Author Response · Authors · 2025-11-25
>
> > Although the authors state that the approach improves performance on benchmarks, I don't think they compared to a baseline-- eg, what if you just did the training but didn't have the attention to the original question blocked?
>
> We have now run this comparison across all six datasets. In Table 2 we report a Non-Markovian GRPO baseline that uses the same RL setup but allows the answer to attend to (Question + CoT). Markovian training tracks this baseline within a few percentage points on average, while enforcing the stricter bottleneck and the interpretability properties studied in the paper.
>
> > It would also be nice to have some qualitative discussions of the chains of thought that result from this training procedure, whether through example transcripts or through some grading of readability.
>
> We have expanded the qualitative discussion. The appendix previously contained examples of arithmetic and Wikipedia continuation transcripts, and we have now added a GSM8K transcript in order to illustrate how trained CoTs differ from the baseline in structure and readability.
>
> > I'm not sure how to interpret figure 3. I'm assuming the x-axis ('sample') is number of tokens sample. Are there some examples of what these chain of thought transcripts look like? Are there parts of the transcript where the llama model has stated the answer?
>
> In Figure 3, the x-axis is the training sample index (batches over time), not token count. The goal is to show how the reward signals from different models co-evolve as training progresses. Rather than plotting a single global correlation (which would be dominated by variation in question difficulty), we show smoothed reward trajectories so that long-range correlations between "LLaMA thinks this is a good CoT" and "other models also reward this CoT" are visible over training.
>
> > I also would like more clarification to interpret the perturbation analyses. Is this analysis testing how well the model adheres to the exact continuation of the wikipedia text? If so, why not do the same perturbation analyses on the previous QA tasks?
>
> The Wikipedia perturbation analyses are with respect to the log-probabilities of the entire next chunk of text (100-token continuation), so they measure how much the CoT affects the model’s ability to match the true future text. In response to your suggestion, we have added perturbation analyses for QA tasks in Appendix B using exact answer accuracy. The effect there is weaker (on the order of a 5-point accuracy drop on average), which is expected because for multiple-choice QA there are many local ways to nudge the correct option’s probability without changing the full reasoning trace.

---

### Official Review · Reviewer_w9RC · 2025-11-01

**Soundness:** 3
**Presentation:** 2
**Contribution:** 2
**Rating:** 4
**Confidence:** 3

**Summary:**

The paper introduces a Markovian Language Model (MLM) framework that enforces structural constraints on Chain-of-Thought (CoT) reasoning. The key insight is treating CoT as a "reasoning autoencoder" where the model must compress essential information into interpretable text that serves as a bottleneck between question and answer. By preventing the model from accessing the original question during answer generation (only seeing the CoT), the framework forces causal reliance on the CoT.

**Strengths:**

It is a nice and elegant idea to introduce causal reliance by construction. The distinction between "faithfulness" and "informativeness" is pragmatic and operationalizable. The reinforcement learning formulation seems sound and there is good improvement after training. There is interesting cross-model generalization where learned CoTs in one architecture transfer to another architectures

**Weaknesses:**

The paper oscillates between two different stories: compression (Wikipedia experiment) and sufficiency (for QA experiment).

The experiments are a bit superficial, there is only one LLaMA model and no baselines such as SFT and GRPO (yet the appendix F anyway shows some Wikipedia results for other models, then why not report results on the QA dataset as well?).
It would be necessary to compare to other post-training baselines, especially on these datasets where any type of reinforcement learning.
Furthermore, It is not clear what is the training data for the experiments in section 5.1, and there is no notion of uncertainty that is being reported (like std or confidence intervals). Appendix D.1 further discuss alternative reinforcement learning formulation, but it is not clear how and why the one in the main paper was chosen.

**Questions:**

How does the approach compare to baselines finetuning methods on QA dataset?
What was the training data in QA experiments?

---

> ### Author Response · Authors · 2025-11-25
>
> > The paper oscillates between two different stories: compression (Wikipedia experiment) and sufficiency (for QA experiment).
>
> These two stories are closely related, and your comment helped us clarify the connection. In the revision we frame both under a single explanation-theoretic view: a good CoT $B$ for a question-answer pair $(A,C)$ is one that is cheap to compute from $A$, makes $C$ cheap to compute given $B$, and has low description length itself. We formalize this using a Levin-style resource-bounded complexity lens and use it to reinterpret both the Wikipedia "compression" setting and the QA
> "sufficiency" setting under a unified notion of good explanations.
>
> > The experiments are a bit superficial, there is only one LLaMA model and no baselines such as SFT and GRPO (yet the appendix F anyway shows some Wikipedia results for other models, then why not report results on the QA dataset as well?).
>
> With respect to "only one LLaMA model," the original submission already included multi-model results on Wikipedia (LLaMA, Phi, Qwen3, and Mistral in Fig. 5a/b), but those were limited to continuation. In the revision we additionally train Qwen3 on each of our six datasets and report these QA results in Appendix C.1 (Table 4).
>
> > It would be necessary to compare to other post-training baselines, especially on these datasets where any type of reinforcement learning.
>
> With respect to SFT and GRPO, our method is in fact a GRPO-style method. The main novelty is the Markovian format/architecture, and our optimization builds on GRPO with two additions: actor-reward gradients and a frozen CoT baseline. To address the concern about post-training baselines, Sec. 6.1 now includes Expert Iteration (STaR-style) and Policy Gradient/EMA ablations; across QA datasets our Markovian GRPO consistently outperforms these, while the ``Baseline'' columns correspond to standard SFT on the instruction-tuned models.
>
> > Furthermore, It is not clear what is the training data for the experiments in section 5.1, and there is no notion of uncertainty that is being reported (like std or confidence intervals). Appendix D.1 further discuss alternative reinforcement learning formulation, but it is not clear how and why the one in the main paper was chosen.
>
> The training data in Sec. 5.1 are the standard training splits of each benchmark (GSM8K, MMLU, SVAMP, ARC-Challenge, Arithmetic, and Wikipedia continuation), and we evaluate on their corresponding test sets; we now clarify this in the experimental setup. We have also added a table of bootstrap confidence intervals in the appendix (Table 5) for each dataset and method ablation. Appendix E.1 describes alternate RL formulations (PG, EI, GRPO+actor-reward), and we chose the GRPO+actor--reward variant as our main method because it was empirically the most stable and highest-performing across tasks (see Sec. 6.1 and the new ablation table).
>
> > How does the approach compare to baselines finetuning methods on QA dataset? What was the training data in QA experiments?
>
> When you say finetuning methods we interpret this as SFT or CoT-distillation baselines. In our tables, the frozen instruction-tuned LLaMA and Qwen3 checkpoints serve as SFT baselines, and our EI configuration approximates a STaR-style CoT distillation baseline that does not rely on a larger teacher model. Across QA datasets, Markovian GRPO improves substantially over both the SFT baselines and EI; we have clarified this comparison in Sec. 6.1.

---

### Official Review · Reviewer_Rwtw · 2025-11-04

**Soundness:** 3
**Presentation:** 3
**Contribution:** 2
**Rating:** 4
**Confidence:** 4

**Summary:**

The paper proposes a Markovian Language Model (MLM) framework that enforces a causal dependency between a model’s Chain-of-Thought (CoT) and its final prediction. The key idea is a “reasoning autoencoder” architecture that introduces a text-based bottleneck: the model must first generate a CoT, and only that CoT (not the original question) is used to produce the answer. The model is trained using a GRPO-style policy gradient algorithm, where the reward depends on how informative the generated CoT is for answer prediction. Empirical results show large improvements on reasoning benchmarks and higher perturbation sensitivity to CoT edits, suggesting the CoTs have become ''load-bearing.''

**Strengths:**

- The idea of enforcing a Markovian structure to make CoTs causally essential is novel, conceptually elegant, and well-motivated.

- The introduction of informativeness as a learning objective is interesting and moves beyond traditional notions of faithfulness or interpretability.

- The formalisation of the Markovian LM and integration of actor–reward gradients (where the reward depends on the same model parameters) are technically sound and well-presented.

- Empirical results show strong and consistent improvements on reasoning benchmarks.

- The perturbation sensitivity and cross-model transfer analyses go beyond accuracy metrics, probing whether the model is actually using CoTs.

**Weaknesses:**

- It is unclear which components of the method (Markovian bottleneck, actor–reward coupling, within-batch normalisation, or reward design) are responsible for the observed gains. The paper should include controlled ablations to isolate these effects.

- The informativeness criterion works well for deductive or mathematical reasoning tasks where the CoT captures logical steps. However, for non-deductive or knowledge-grounded tasks (e.g., MMLU, factual QA), informativeness alone may be insufficient.  Additionally, the authors claim that forcing the model to predict the answer only from CoT improves informativeness. Still, there is no baseline from which the model predicts (Question + CoT) under the same RL training. This comparison is critical to verify that the performance gains truly stem from the Markovian constraint.

- There is no comparison against recent process-supervised or fine-tuned CoT methods such as STaR. This makes it difficult to judge whether the proposed Markovian constraint provides a real advantage over established reasoning-enhancement techniques.

**Questions:**

Missing citations:

1. Making Reasoning Matter: Measuring and Improving Faithfulness of Chain-of-Thought Reasoning EMNLP Paul et.al. 2024
2. Truthful or Fabricated? Using Causal Attribution to Mitigate Reward Hacking in Explanations ICML Ferreira et. al. 2025

---

> ### Author Response · Authors · 2025-11-25
>
> >  It is unclear which components of the method (Markovian bottleneck, actor–reward coupling, within-batch normalisation, or reward design) are responsible for the observed gains. The paper should include controlled ablations to isolate these effects.
>
> We have added Table 2 in the main paper with selected ablations and a more extensive version (including full sweeps and confidence intervals) in Table 5 in the appendix. These cover ablations for the Markovian bottleneck, actor–reward coupling, the reward design (the “Unnorm” variant), and a variant that replaces GRPO-style within-batch standardisation and parallel sampling with an exponential moving-average (EMA) baseline. We find that the Markovian bottleneck only causes a few percentage points drop across tasks.
>
> > The informativeness criterion works well for deductive or mathematical reasoning tasks where the CoT captures logical steps. However, for non-deductive or knowledge-grounded tasks (e.g., MMLU, factual QA), informativeness alone may be insufficient. Additionally, the authors claim that forcing the model to predict the answer only from CoT improves informativeness. Still, there is no baseline from which the model predicts (Question + CoT) under the same RL training. This comparison is critical to verify that the performance gains truly stem from the Markovian constraint.
>
> Wikipedia text prediction is non-deductive, and we show successful training of this across 4 models and 4 training runs on the same model in Figures 5a and 5b. In addition, we now include a Non-Markovian GRPO baseline that predicts from (Question + CoT) under the same RL setup, so the reviewer’s requested comparison is present: it serves as an upper bound, and the Markovian model tracks it closely while offering the stronger interpretability guarantee.
>
> > There is no comparison against recent process-supervised or fine-tuned CoT methods such as STaR. This makes it difficult to judge whether the proposed Markovian constraint provides a real advantage over established reasoning-enhancement techniques.
>
> GRPO for CoT training (DeepSeek-Math) is more recent than STaR, but we also added an Expert Iteration (EI) baseline which is closer to the spirit of STaR: it keeps only high-reward CoTs and trains on those. EI underperforms our full Markovian GRPO recipe across datasets, which we now point out explicitly.
>
> > Missing citations
>
> We've added and discussed references to both of these papers, thanks for bringing them to our attention.

---

### Author Response · Authors · 2025-11-25

The general sentiment of the reviews is that this paper is novel, important, and has strong results, but that it is missing three main aspects:
i) a comparison to the performance of a model trained with the question still in the context window, in terms of answer-prediction performance
ii) ablations clarifying why each of the design decisions for optimizing the Markovian objective was made
iii) a clear motivation for the Markovian framing that does not oscillate between compression and sufficiency.

**Points (i) and (ii)**

We have addressed (i) and (ii) by running a full sweep of training runs over Markovian vs.\ Non-Markovian variants and four different method ablations on six datasets, with uncertainty estimates (bootstrap confidence intervals) reported in Table 5. We find that our chosen set of techniques provides significant performance benefits over their ablations. Rerunning our primary training run for more RL steps (10,000 datapoints) has also improved our headline performance numbers for Llama (GSM8K 57.1\%, ARC-Challenge 79.9\%, MMLU 55.5\%). We additionally include Table 4, showing that Qwen reaches high performance on several datasets (GSM8K 71.6\%, ARC-Chal 85\%, MMLU 60.5\%) using the same training algorithm and hyperparameters.

In comparison to training with access to the question during prediction (Non-Markovian), we only lose about 3-4 percentage points on average. Non-Markovian is meant to be an upper bound, since it has access to strictly more information during training and evaluation. In exchange, we obtain the interpretability benefits studied in Sections 5.2 through 5.4: answer prediction becomes increasingly fragile with respect to changes in the CoT, showing that the particular CoT was load-bearing for prediction. We also find evidence that our CoTs do not consist of model-specific steganography, since a collection of diverse models is able to predict the answers given the learned CoTs.

This explicit framing of the trade-off—interpretability benefits in exchange for only a small performance cost—is a clarification to the presentation of the paper that we owe to this review process. Otherwise, the paper might give the impression that we were trying to beat Non-Markovian on prediction performance full-stop, which would be suspect.

Lastly, reviewer v7a8 asked why we do not run perturbation experiments on QA tasks, so we added those as well in Table 3.

**Point (iii)**

Reviewer w9rc pointed out:
> The paper oscillates between two different stories: compression (Wikipedia experiment) and sufficiency (for QA experiment).

We thank the reviewer for bringing this point to us. The original version did oscillate in a way that made the presentation unclear, reflecting a real underlying tension that we have tried to resolve more cleanly with the new "Algorithmic view of explanations" paragraph in the introduction (and corresponding clarifications elsewhere).
We introduce CoT fragility as important because it is really a proxy for compression. But compression of what?
If we were compressing only the answer, that might work as an explanation for the Wikipedia example, which has long subsequent text to predict; but for question--answer datasets (especially multiple choice) the answer string itself is often very short, so a purely "compress the answer" story is unsatisfying.
The takeaway is that this paper is really about learning to generate good explanations. A good explanation $B$ of a string $C$ given a string $A$ is one which is (i) easy to generate from $A$, (ii) helpful for predicting $C$, and (iii) simple. Heuristically, this connects to the quantity $K_t(B\mid A) + K_t(C\mid B) + K_t(B)$, where $K_t$ is Levin time-bounded complexity. In the paper we stick to informal, MDL-style language and avoid heavy formalism, but this explanation-theoretic lens helps reconcile the "compression" and "sufficiency" perspectives.
This fixes the QA-pair counterexample, because even though an answer may be short, it may be arbitrarily difficult to predict; in our setting, that difficulty is reflected in its effective code length, i.e., $-\log \pi(\text{ans} \mid \text{CoT})$.

---

### Meta-Review · Area_Chair_6TsA · 2026-01-11

**Summary:**

This paper investigates how to make the thoughts in CoT more informative by encouraging them to explicitly compress the information necessary to solve a problem, such that the thoughts alone are sufficient to derive the answer. To this end, the authors propose a Markov transformer framework that substantially improves the predictability of the generated thoughts.

Although the proposed approach underperforms standard RL-based CoT training in terms of end-task accuracy, I believe this work constitutes an important step toward understanding the role of “thoughts” in recent thinking models. Moreover, the ability to reliably compress useful information into interpretable intermediate representations may have meaningful implications for future research.

**Reviewer Concerns:**

In my view, all reviewer concerns have been adequately addressed. I understand that reviewers requesting comparisons with standard RL training may remain dissatisfied with the observed performance gap. However, given that this work represents an early investigation in this direction, I believe this limitation is acceptable.

**Reviewer Scores:**

As discussed above, the two reviewers may not increase their scores. However, I believe it would be fair to say that the author responses have adequately addressed their concerns.

---

### Decision · Program_Chairs · 2026-01-26

Accept (Poster)